# Autumn Film Mulched Ridge Microfurrow Planting Improves Yield and Nutrient-Use Efficiency of Potatoes in Dryland Farming

**Fengke Yang** [1,2,3,4,*]**, Baolin He** [1,2,3]**, Bo Dong** [1,2,3,4] **and Guoping Zhang** [1,2]

1 Dryland Farming Institute, Gansu Academy of Agricultural Sciences, Lanzhou 730070, China; blihe@163.com (B.H.); dongbobby@163.com (B.D.); zhangguoping79@126.com (G.Z.)
2 Key Laboratory of Efficient Utilization of Water in Dryland Farming of Gansu Province, Lanzhou 730070, China
3 Key Laboratory of Low-Carbon Green Agriculture in Northwestern China, Ministry of Agriculture and Rural Affairs, Yangling 712100, China
4 The Joint Key Laboratory of Ministry of Agriculture and Rural Affairs-Gansu Province for Crop Drought Resistance, Yield Increment and Rainwater Efficient Utilization on Rain-Fed Area, Lanzhou 730070, China
* Correspondence: yang_fk@163.com

**Abstract:** Potatoes (*Solanum tuberosum*) are the most important noncereal crop in the world. Increasing potato production is critical for future global food security. China is the world's largest potato producer, and potato productivity is constrained by water scarcity and poor fertilizer use efficiency (NUE$_F$). Recently, autumn film mulched ridge microfurrow rainwater harvesting (ARF) tillage has been successfully applied in potato production in dryland farming in Northwest China. However, the effects of ARF on the use efficiency (NUE$_F$) of applied nitrogen (N), phosphorus (P), and potassium (K) nutrients in potatoes have not been systematically studied. A 3-year, consecutive field trial with four treatments, including moldboard planting without fertilizer application (control, CK), spring and autumn film mulched ridge microfurrow rainwater harvesting planting (SRF and ARF), and standard film mulched ridge-furrow planting (FRF), was conducted during 2018–2020. ARF greatly increased the water levels in the 0–200 cm soil profile at potato harvest compared to SRF, FRF, and CK. ARF and SRF significantly increased the levels of soil organic carbon (SOC), total and available NPK (TN, TP, TK and AN, AP, AK) compared to FRF and CK, with ARF being the most efficient at increasing the levels of the AN, AP, and AK. ARF significantly improved the soil water and nutrient activity and contributed the most to potato tuber and biomass yield and hence the NUE$_F$. Under ARF, significant and positive associations were observed between the soil fertility traits, soil water storage (SWS), potato tuber yield, biomass yield, and NUE$_F$. Soil fertility traits and the SWS were positively correlated with potato tuber and biomass yield. The SWS, potato tuber, and biomass yield positively correlated with the partial factor productivity (PFP) and the recovery efficiency (RE) of the applied NPK nutrients. Increased nutrient levels and their combination increased the NUEF and NUEF's components. The TN and AN contributed more significantly to the PEP and agronomic efficiency (AE) of the applied NPK nutrients; the TP was significantly positively correlated with the AE (AE$_N$, AE$_P$, and AE$_K$), while the AP was correlated with PEP (PFP$_N$, PFP$_P$, and PFP$_K$) and RE (RE$_N$, RE$_P$, and RE$_K$); the TK was significantly positively correlated with the PFP and RE of the applied PK nutrients, while the AK was significantly positively correlated with the PE$_P$, AE, and RE of the applied K nutrients. Therefore, ARF results in a synchronous increase in yield and NUE$_F$ and is the most efficient planting system for potato production in dryland farming.

**Keywords:** autumn film mulched ridge microfurrow rainwater harvesting; soil fertility; soil nutrient-use efficiency; potato; dryland farming





## 1. Introduction

Potato (*Solanum tuberosum* L.), the third most consumed [1,2] and fourth most cultivated crop worldwide [3–5], is crucial for future global food security [6,7]. China is the

world's leading potato-growing nation; its potato harvest area is 578.3 million hectares, which accounts for 31.9% of the world's total harvest area (1813.3 million ha), while the average yield is 16,317.9 kg·ha$^{-1}$, which is 78.7% of the world average (20,742.6 kg·ha$^{-1}$) [8]. Moreover, approximately one-third of potato production occurs in the northwestern provinces (Inner Mongolia and Gansu) [9]; however, the yield here is below the world average [10–13]. In addition, high rates of chemical fertilizer application, imbalanced nutrient applications with low nutrient use, high fertilizer costs, and environmental problems have highlighted the need to increase tuber yield [14–16] and the nutrient-use efficiency of applied nitrogen, phosphorus, and potassium fertilizer (NUE$_F$) synchronously [17–20]. Estimating the total NUE$_F$ of the current potato production system is important to increase potato yield and ensure food security.

NUE$_F$ is specifically used to separate from nitrogen-use efficiency (NUE) to avoid confusion with NUE, which was used in previously published articles. Although potatoes can be highly productive, their relatively shallow–sparse root system [21–23] that limits nutrient uptake and utilization [23,24] results in a lower NUE$_F$ compared to other crops [4,10,25–27]. Autumn film mulched ridge microfurrow rainwater harvesting (ARF) planting, which is a combination of film mulching and ridge-furrow rainwater harvesting tillage, has been widely adopted for potato production in Northwest China [28,29]. ARF integrates the use of high-yielding and nutrient-efficient varieties [10,19,30–33], 4R (right source, right rate, right time, and right place) best practices for fertilization management [34,35], the effective regulation of soil water fertility conditions [36], and major benefits for improving the NUE$_F$ [18,36–38]. Studies have shown that ARF produces the highest tuber yield and good potato quality, mainly because of the significantly increased potato water-use efficiency [11,39,40]; however, whether it increases the NUE$_F$ is largely unknown.

The NUE$_F$ is the yield per unit of nutrient resources available to plants [41]. Soil fertility, especially soil macronutrient (nitrogen (N), phosphorus (P), and potassium (K) availability, drives optimal plant production and NUE$_F$. NUE$_F$ has been reported to be associated with the gain and loss of soil organic carbon (SOC), which is closely related to the gain and loss of key growth-limiting nutrients, including NPK [42,43]. SOC and NPK have been instrumental in determining the availability of soil nutrients that are strongly associated with potato yield and NUE [44]. N, P, and K are the built-in components of the energy-rich phosphate compounds adenosine diphosphate (ADP) and adenosine triphosphate (ATP), which play a key role in various physiological and metabolic mechanisms [45]. N is the primary component of chlorophyll, nucleic acids, proteins, amino acids, coenzymes, and membrane constituents [46]. Increasing the N-use efficiency can increase vegetative growth [35] and the proportion of large tubers [10,17,47,48]. P is involved in energy transfer, photosynthesis and respiration, rapid canopy development, root cell division, tuber set, and starch synthesis [17,49–51]. Increased P-use efficiency is essential to hasten maturity, induce plants to tolerate drought and some diseases [52], and optimize tuber yield [35]. K is vital in photosynthesis, enzyme activation, photosynthate translocation, turgidity maintenance, and osmoregulation [51,53]. Increasing K use increases the yield and yield components and impacts the tuber quality [54]. P and K also stimulate root growth and formation, which is particularly important for potato plant water and nutrient uptake [17,31,44]. Furthermore, SOC, N, P, and K interact positively with each other and regulate the uptake of other nutrients [17] and soil carbon metabolism [55], which influences plant nutrition, crop yield, and NUE$_F$ [22,56]. Previous studies on the NUE$_F$ mainly focused on the use efficiency of N [57–59] and paid little attention to P [32,44,47,52,60–63] and ignored K [17] based on a notion of sufficient soil K in potato production in Northwestern China [62]. Therefore, systematically assessing the NUE$_F$ to scientifically evaluate ARF planting systems is necessary.

Previous studies have often used partial factor productivity (PFP) and agronomic efficiency (AE) to assess the nitrogen-use efficiency (NUE) of potatoes but have not fully explained the NUE$_F$. Therefore, this study assessed the underlying mechanisms by which ARF increases the NUE$_F$ and potato productivity using the PFP, AE, recovery efficiency (RE),

internal use efficiency (IE), and reciprocal IE (RIE). We hypothesize that besides improving the soil water status, ARF improves soil fertility and increases nutrient availability, which results in a high $NUE_F$ and yield. Using treatment with moldboard planting without fertilizer application as a control (CK), we systematically compared the effects of ARF versus spring film mulch microfurrows rainwater harvesting planting (SRF) and standard film mulch ridge-furrow (FRF) planting on the $NUE_F$ and potato yield. The specific objectives of the study were (1) to determine the effect of ARF compared with those of CK, SRF, and FRF on SOC, N, P, and K nutrient contents and their availability and potato yield; (2) to characterize the N, P, and K nutrient-use efficiency under ARF; and (3) to investigate the possible mechanism by which ARF improves the $NUE_F$ and potato yield.

## 2. Materials and Methods

### 2.1. Site Description and Experimental Design

A field experiment was conducted using a potato (cv. Longshu 7) at the Zhuanglang Testing Station of the Gansu Academy of Agricultural Sciences (106°05′28″ E, 35°10′30″ N; 1765 m above sea level), which is a typical potato planting region in Gansu, Northwest China. This region has an average annual temperature of 7.9 °C, a frost-free period of 145 days, and average annual precipitation and evaporation levels of 510.4 and 1289.1 mm, respectively. Overall, 70% of the precipitation occurs between July and September (rainy season), whereas the other months receive less precipitation. The annual precipitation distribution patterns nearly correspond to the water-consuming period of potato growth. During the potato-growing season (20 April to 10 October), the total rainfall was 341.3, 405.0, and 560.8 mm in 2018, 2019, and 2020, respectively, which were considered dry, normal, and wet, respectively, compared with the average (398 mm from 2000–2020). The experimental field had loessial soil [64], which was similar to the Calcaric Cambisols of the FAO/UNESCO soil classification system [65]. The basal physicochemical characteristics of the 0–20 cm soil layers are shown in Table 1. Moldboard plowing (20–25 cm deep) combined with rotary tillage (approximately 30 cm deep) was carried out once a year as this is the region's standard soil tillage regime.

**Table 1.** The basal physicochemical characteristics of the 0–20 cm soil layers for the experimental fields in 2018.

| Soil Layer (cm) | Soil Texture | pH | SOC (g·kg⁻¹) | TN (g·kg⁻¹) | TP (g·kg⁻¹) | TK (g·kg⁻¹) | AN (mg·kg⁻¹) | AP (mg·kg⁻¹) | AK (mg·kg⁻¹) | BD (g·cm⁻³) |
|---|---|---|---|---|---|---|---|---|---|---|
| 0–20 | Clay loam | 8.5 | 9.1 | 0.76 | 0.72 | 17.5 | 66.5 | 15.0 | 176.6 | 1.33 |
| 20–40 | Sandy loam | 8.5 | 8.4 | 0.49 | 0.68 | 13.6 | 58.6 | 13.5 | 130.5 | 1.34 |
| 40–60 | Sandy loam | 8.5 | 6.7 | 0.41 | 0.67 | 12.5 | 52.3 | 10.3 | 121.1 | 1.36 |

Note: SOC, soil organic carbon; TN, total nitrogen; TP, total phosphorus; TK, total potassium; AN, alkali-hydrolyzable nitrogen; AP, available phosphorus; AK, available potassium; BD, soil bulk density.

The field experiment was carried out by using a randomized block design with four treatments and three replicates as follows: (1) CK (control), whereby we used a moldboard plow and planted every two plows with one plow empty, and no mulching or fertilizing practices were performed (Figure 1A); (2) FRF, standard film mulched ridge-furrow planting, whereby the FRF plots were arranged alternately in narrow furrows (15 cm high × 40 cm wide) and wide ridges (10 cm high × 70 cm wide) and were prepared before planting the potatoes in mid-April (Figure 1B); (3) ARF and SRF, autumn and spring film mulched ridge microfurrows rainwater harvesting planting, whereby the plots were alternately arranged in a wide ridge (60 cm wide × 24 cm high) and microfurrows (40 cm wide × 24 cm high) [66] and were prepared every second 10 days in March when the soil started thawing (for SRF) and every second 10 days in October immediately after the rainy season of the last year (for ARF) by using a man-made simple tool (Figure 1C).

All plots in the FRF, ARF, and SRF groups were completely mulched by using a 120 cm wide and 0.0018 cm thick polyethylene film, with a 0.5 cm diameter rainwater infiltration hole drilled in the furrows every 33 cm. Each replicate plot was 40 m$^2$ (5 × 8 m) and was separated by a 0.6 m wide walkway. Potatoes were sown in the plow furrows following tillage at sowing time for CK and on the ridges for FRF, ARF, and SRF. The potato planting density was 50,000 cavesha$^{-1}$ with a row spacing of 60 cm and a cavity spacing of 33 cm (Figure 1).

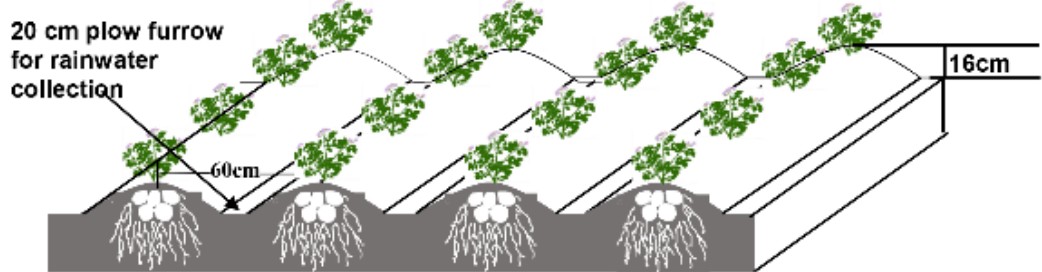

**(A) Traditional moldboard planting without fertilizer application, serve as CK**

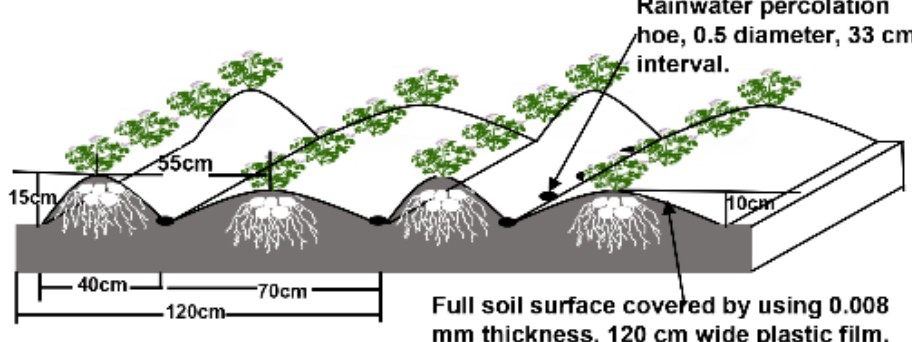

**(B) Full film mulched ridge-furrow rainwater harvesting planting, FRF**

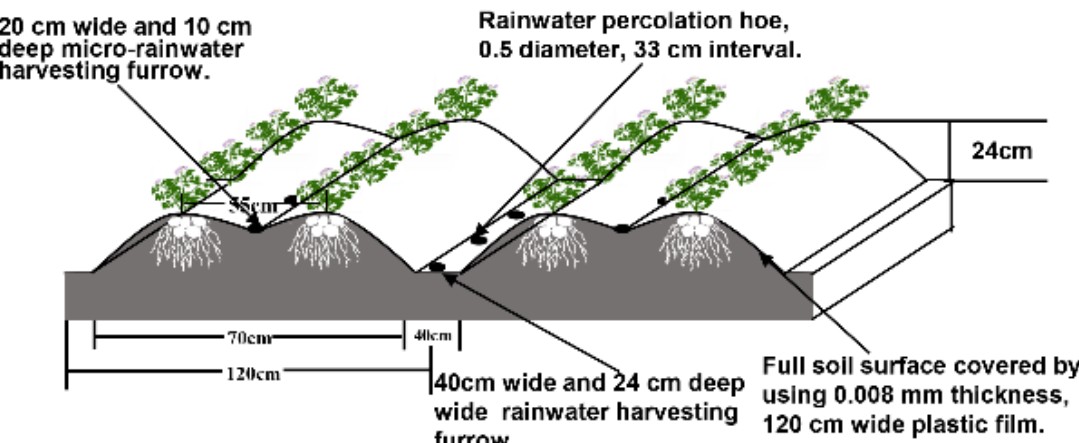

**(C) Autumn or Spring full film mulched micro-furrows rainwater harvesting planting, ARF &SRF**

**Figure 1.** Sketch of film mulched ridge with microfurrow rainwater harvesting planting.

CK, farmer traditional bare land heap-up earth planting; FRF, film mulched ridge-furrow rainwater harvesting planting (standard planting pattern typically used in dryland farming in the Loess Plateau of China); ARF and SRF, autumn and spring full film mulched microfurrow rainwater harvesting planting.

Recommended N 180.0 kg·ha$^{-1}$ doses, P$_2$O$_5$ 90.0 kg·ha$^{-1}$, and K$_2$O 180 kg·ha$^{-1}$ were applied for FRF, ARF, and SRF [9]. The sources of the chemical fertilizers were urea (46% N),

superphosphate (12% $P_2O_5$), and potassium sulphate ($K_2SO_4$) (52% $K_2O$). Additionally, 7.5 t ha$^{-1}$ of farmyard manure in pig slurry (SOC 20–25 g·kg$^{-1}$, N 1.5–2.0 g·kg$^{-1}$, $P_2O_5$ 0.80–2.50 g·kg$^{-1}$, AN 180–225 mg·kg$^{-1}$, AP 12.9–17.2 mg·kg$^{-1}$, and AK 232–291 mg·kg$^{-1}$) was applied as organic fertilizer for FRF, ARF, and SRF. Half of the N fertilizer, the total P and K fertilizers, and the farmyard manure were used as the base fertilizer in each plot. The other half of the N fertilizer was applied at the potato tuber initiation stage at a depth of 15 cm between the plants by using hand-held implements.

### 2.2. Soil Sampling and Physicochemical Property Analysis

### 2.2.1. Soil Water Content

The soil water content was gravimetrically determined by using a soil auger to collect the samples manually at a depth of 200 cm at 20 cm intervals in each plot before sowing and after harvest each year. Five soil samples were collected from each plot by using the w-pattern sampling method at each sampling time. The sampling sites were located at distances equal to half of the furrow width from the planting furrows, which spanned from the boundaries of the ridges and furrows as described by [67]. Fresh soil samples in aluminum boxes were weighed first, then oven dried at 105 °C for 48 h, and weighed again to estimate the soil water content [39].

### 2.2.2. Soil Nutrients Content

Soil samples were collected after potato harvesting in early October each year. A total of 36 cores, 3 in each plot at 0–20, 20–40, and 40–60 cm, were randomly collected by using an auger (5 cm diameter), with three cores in each plot. Sampling was conducted from the middle of the planting furrows (wide ridges) and the middle of the narrow furrows between two plants. These 36 samples were bulked, homogenized, and sieved through a 2 mm mesh to remove stones and surface litter for soil fertility parameter testing.

The SOC was measured by using a multi N/C$^®$ 3100 TOC analyzer (Analytik Jena, Jena, Germany), and total nitrogen (TN) was estimated following the $KMnO_4$ oxidation method. Alkali-hydrolyzable nitrogen (AN) was determined by using the alkalizable diffusion method. The available phosphorus (AP) was determined by using the Olsen sodium bicarbonate method, and the total phosphorus (TP) was measured by using a colorimetric test after digestion with an $HClO_4$-$H_2SO_4$ mixture. The total potassium (TK) was estimated by using dissolution-flame photometry, and the available potassium was estimated by using the ammonium acetate ($NH_4Oac$) method.

### 2.3. Potato Tuber Yield, Yield Component, Root, and Total Biomass Yield Measurement

At harvest each year, potato plants in the central 30 m$^2$ were harvested from a 40 m$^2$ area in each plot to determine the tuber yield. Ten plants were selected from the middle two rows of each plot to determine the tuber number, tuber weight, haulm, and root weight per plant, and tuber rates were categorized according to the following grades: large (>150 g), medium (150–75 g), and small (<75 g). The potato tuber yield (kg·ha$^{-1}$) was determined at 85% moisture content [13] and calculated as the plot yield (kg) divided by the plot area (m$^2$) and then multiplied by 10,000. The haulm and root weight per ha, which were also at an 85% moisture content, were calculated as their average weight per plant multiplied by the planting density (50,000 plants ha$^{-1}$). Then, the total biomass (kg·ha$^{-1}$) was determined as the sum of the tuber, haulm, and root yield per ha [13,39,68].

### 2.4. Plant Nutrient Content Analysis

Three plants per plot were sampled and separated into tubers, haulms, and roots. The samples were fixed at 105 °C for 30 min and dried to a constant weight at 65 °C. The dry weights of the tubers and haulms were recorded. The samples for the nutrient content tests were then crushed separately and evenly mixed. Briefly, the plant samples were digested with $H_2SO_4$-$H_2O_2$, nitrogen was determined by using the Kjeldahl method, phosphorus

was determined by using the phosphorus vanadium molybdate yellow colorimetric method, and potassium was determined by using the flame photometric method [67,68].

*2.5. Water-Use Efficiency (WUE) and NUE$_F$ Calculation*

2.5.1. WUE

The WUE (kg·ha$^{-1}$ mm$^{-1}$) was calculated as Y/ET. Y is the tuber yield (kg·ha$^{-1}$), and ET is the evapotranspiration (mm) calculated by ET = P + ΔW, whereby P is the rainfall and ΔW is the difference in the water storage in the 0–200 cm soil profile between the planting and harvesting period [67]. Meanwhile, the water storage of the soil profile from 0–200 cm (SWS, mm ha$^{-1}$) at harvest every year was calculated by SWS = ρ × h × W × 10, where h is the soil depth (cm), ρ is the soil bulk density (1.30 g cm$^{-3}$), b is the gravimetric soil moisture content (%), and 10 is the unit conversion factor [67].

2.5.2. NUE$_F$

According to the definition of nutrient-use efficiency [41,69] commonly used in the literature [60,62,70], we used the PFP (partial factor productivity, kg of tuber yield produced per unit of nutrient applied), AE (agronomy efficiency, kg of tuber yield increase per unit of nutrient applied), apparent RE, (recovery efficiency, nutrient uptake (tuber + haulm) increase per unit of nutrient applied), and IE (internal use efficiency, yield (tuber + haulm) per unit of nutrient applied) to characterize the NUE$_F$ of the potato. These were calculated by using the following formulas [12,47,71]:

$$\text{PEP for N; P or K} = \frac{tuber\ yield}{amount\ of\ N;\ P_2O_5\ or\ K_2O\ fertilizer\ applied} \tag{1}$$

$$\text{AE for N; P or K} = \frac{tuber\ yield\ in\ NPK\ plots\ -\ tuber\ yield\ in\ N;\ P\ or\ K\ empty\ control\ plots}{amount\ of\ N;\ P_2O_5\ or\ K_2O\ fertilizer\ applied} \tag{2}$$

$$\text{RE for N; P or K} = \frac{[total\ nutrient\ uptake(tuber\ +\ haulm)\ in\ NPK\ plots\ -\ nutrient\ uptake\ (tuber\ +\ haulm)\ in\ N; P\ or\ K\ empty\ control\ plots]}{amount\ of\ N;\ P_2O_5\ or\ K_2O\ fertilizer\ applied} \tag{3}$$

$$\text{IE for N; P or K} = \frac{yield\ (tuber\ +\ haulm)\ in\ NPK\ plots}{amount\ of\ N;\ P_2O_5\ or\ K_2O\ applied} \tag{4}$$

In addition, the N, P, and K nutrient uptake by tons of tubers, i.e., the reciprocal internal use efficiency (RIE), was calculated as follows:

$$\text{RIE for N; P or K} = \frac{1000}{IE} \tag{5}$$

*2.6. Statistical Analysis*

An analysis of variance was performed by using SPSS software (version 19.0; SPSS Inc., Chicago, IL, USA) to assess the effect of different planting regimes on potato tuber, total biomass and root yield, tuber yield components, and selected soil chemical properties and NUE$_F$ indices. Tukey's LSD test was used to compare the treatment means, and the differences were considered significant at $p < 0.05$. In addition, Pearson's correlation analysis was performed to determine the relationship between the soil NPK nutrients content, SWS, potato tubers, biomass yield, and the NUE indices by using three-year average data. Figures were created by using SigmaPlot 14 (Systat Software, Inc., Chicago, IL, USA).

## 3. Results

*3.1. Soil Water and Nutrient*

The ARF, SRF, and FRF uniformly increased the soil water content of the soil profile from 0–200 cm compared to CK at potato harvest, with soil water levels under ARF being higher than that under SRF and FRF during the three test years (Figure 2). The only

difference is that the soil water levels rose with increasing precipitation and was higher in 2020 than in 2019 and 2018 (Figure 2).

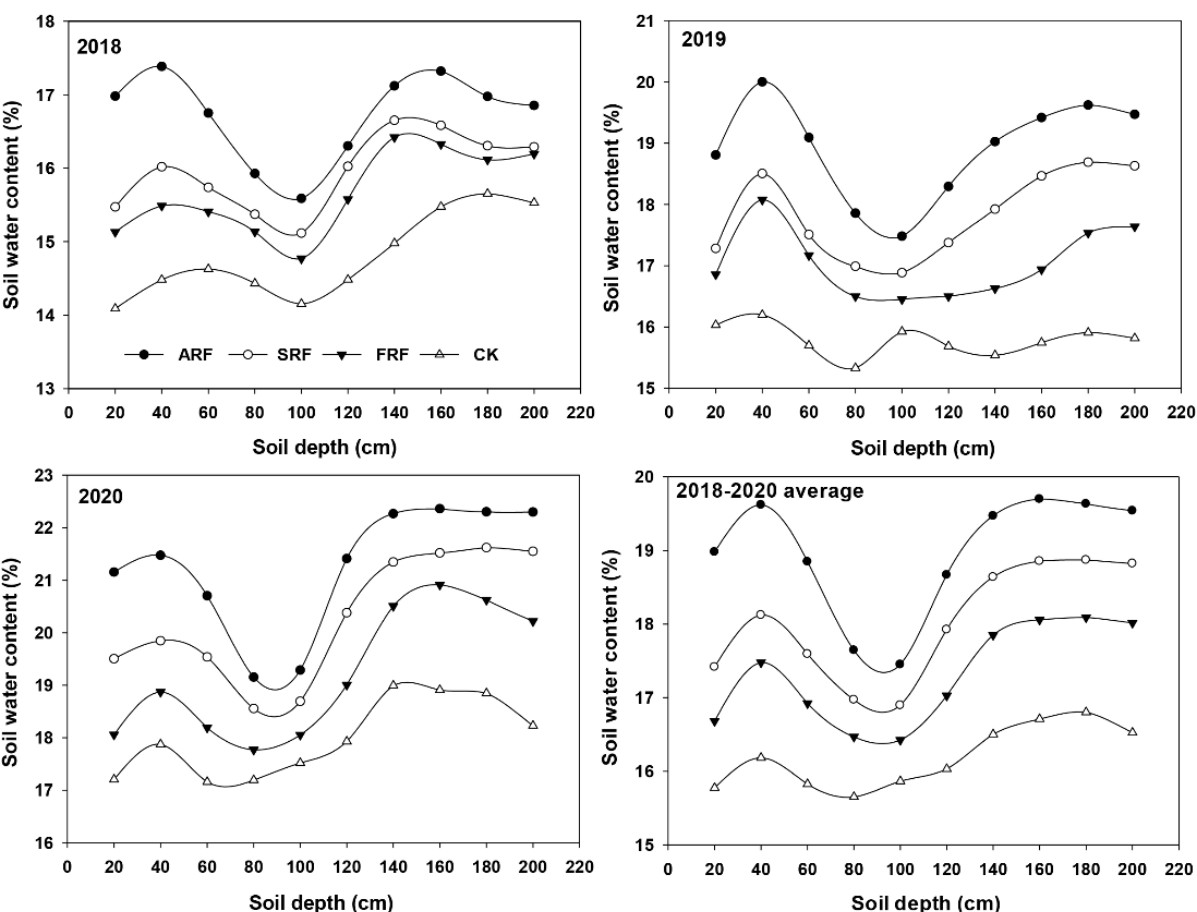

**Figure 2.** Soil water levels in the 0–200 cm soil profile at potato harvest during 2018–2020. CK, farmer traditional bare land heap-up earth planting; FRF, film mulched ridge-furrow rainwater harvesting planting (standard planting pattern typically used in dryland farming in the Loess Plateau of China); ARF and SRF, autumn and spring full film mulched microfurrow rainwater harvesting planting.

ARF and SRF consistently significantly increased the levels of the total soil organic C, N, P, and K (SOC, TN, TP, and TK) in the soil from the 0 to 60 cm soil profile compared with FRF and CK ($p < 0.05$) independent of the precipitation year (Table 2). The values of SOC and TP increased the most in the 0–20 cm soil layer, and those of the TN increased more in the 20–40 cm soil layer than in the 40–60 and 0–20 cm soil layers, whereas the TK values in the three soil layers increased almost equally. Meanwhile, ARF and SRF significantly increased the content of labile organic carbon (LOC) and available phosphorus (AP) levels compared to FRF and CK, and ARF significantly increased the available N and K levels compared to SRF, FRF, and CK consistently in the three soil layers and three test years (Table 2, $p < 0.05$). The greatly improved LOC, especially the AN, AP, and AK, indicated a significant increase in the nutrient availability, which contributed more to the increase in soil fertility.

### 3.2. Potato Tuber, Root, Biomass Yield, and Water-Use Efficiency (WUE)

ARF, SRF, and FRF all increased the potato tubers, roots, total biomass yield, and WUE significantly ($p < 0.05$) compared with CK (Table 3). The values of the four parameters increased with increasing precipitation from the dry year (2018) to the normal (2019) and wet years (2020). The increased tuber rate in the large- and medium-sized tubers and the corresponding weight contributed the most to the increase in the tuber yield. The

yield-enhancing effects of ARF were not significantly different from those of SRF in dry and normal years but were significantly higher than those of SRF in the wet year; both were significantly higher than those of FRF and CK. Regarding the total biomass and fresh root yield, the enhancing effect of ARF was not significantly different from SRF in the dry year but was significantly higher than that of SRF and FRF in the normal and wet years. The WUE was consistently significantly higher under ARF and SRF than under FRF and CK over the three years of testing. On average, ARF increased the potato tuber, root, and biomass yield significantly and improved the tuber quality compared to the other three systems (Table 3).

*3.3. Nutrient-Use Efficiency of Applied NPK Fertilizers*

3.3.1. PFP and AE

ARF, SRF, and FRF uniformly significantly increased the PFP and AE of the NPK fertilizer ($PFP_N$, $PFP_P$, $PFP_K$, $AE_N$, $AE_P$, and $AE_K$) compared with those of CK (Table 4, $p < 0.05$). The PFP values of the NPK fertilizer were the highest in the normal year, moderately higher in the wet year, and lower in the dry year. The $PFP_P$ was higher than the $PFP_K$ and $PFP_N$. There was no significant difference between the ARF and SRF in the $PFP_N$, $PFP_P$, and $PFP_K$ in the dry and normal years, but significant differences were observed in the wet years; both had higher PFPs than those of the FRF and CK. The $AE_P$ was also higher than the $AE_K$ and $AE_N$, with the AE values of the NPK fertilizer increasing with increasing rainfall from dry to normal and wet years. The effects of ARF and SRF on the $AE_N$, $AE_P$, and $AE_K$ did not differ significantly from each other and were higher than those of FRF and CK in the precipitation years and the mean precipitation year.

3.3.2. RE, IE, and RIE

ARF, SRF, and FRF had slightly different effects on the RE and no significantly different effects on the IE, RIE, or nutrient uptake per ton of tuber yield of the NPK fertilizers in different rainfall years (Table 4, $p < 0.05$). The values of $RE_N$ and $RE_K$ were almost equal to or higher than those of $RE_P$, and all increased with increasing tuber yield and precipitation (Table 4). Comparing the effects between the three planting systems, ARF significantly increased $RE_N$ and $RE_K$ compared with SRF and FRF in the dry years; ARF and SRF increased $RE_K$ almost equally and significantly compared with FRF in the normal and wet years, and all three had no significant effect on $RE_P$ among the three rainfall years. The values of the $IE_N$, $IE_P$, and $IE_K$ were almost consistent under the three planting systems, with the $IE_P$ being higher than the $IE_N$ and $IE_K$ regardless of the rainfall. ARF and SRF slightly reduced the values of the $IE_K$ compared with FRF. On average, ARF, SRF, and FRF did not significantly affect the $IE_N$ and $IE_P$, whereas ARF significantly decreased the $IE_K$ compared with SRF and FRF. The RIE values demonstrated a trend whereby the $RIE_N$ and $RIE_K$ were almost equal, both higher than the $RIE_P$, and decreased from ARF to SRF and FRF. ARF, SRF, and FRF differed significantly in their effect on the RIE, regardless of the rainfall year. ARF and SRF significantly increased the RIE relative to FRF, on average.

*3.4. Relationship between Soil Water, Soil Nutrient, Potato Tuber, Biomass Yield, and $NUE_F$*

Pearson's correlation analysis was used to determine the contribution of the soil fertility and SWS on the potato tuber yield, biomass yield, and $NUE_F$ under ARF. The total and available soil traits (excluding the TP), as well as the SWS, were significantly positively correlated with the potato biomass and tuber yield (Figure 3A). The potato biomass was significantly positively correlated with the tuber yield. Both were positively significantly correlated with the PFP; the potato tuber yields concurrently contributed more significantly to the $RE_P$ and $RE_K$, while the SWS was most significantly correlated with the PFP and RE (Figure 3B). The TN and AN contributed more significantly to the PEP and AE; the TP was significantly positively correlated with the AE, while the AP was correlated with the PEP and RE; the TK was significantly positively correlated with the PFP, $RE_P$ and $RE_K$, while the AK was significantly positively correlated with the PEP, AE, and $RE_K$ (Figure 3C).

**Table 2.** SOC, NPK content, and their active components in 0–60 cm soil layer in 2018–2020.

| Pa. | Tr. | 0–20 cm | | | 20–40 cm | | | 40–60 cm | | | 0–60 cm Average | | |
|---|---|---|---|---|---|---|---|---|---|---|---|---|---|
| | | 2018 | 2019 | 2020 | 2018 | 2019 | 2020 | 2018 | 2019 | 2020 | 2018 | 2019 | 2020 |
| SOC | CK | 8.9 ± 0.1 b | 9.1 ± 0.1 b | 9.1 ± 0.1 b | 8.2 ± 0.1 b | 8.3 ± 0.2 b | 8.4 ± 0.1 b | 6.6 ± 0.9 b | 6.5 ± 0.9 b | 6.7 ± 0.4 b | 7.9 ± 0.2 b | 7.9 ± 0.2 b | 8.0 ± 0.1 b |
| | FRF | 9.5 ± 0.1 b | 9.5 ± 0.06 b | 9.7 ± 0.1 b | 8.7 ± 0.02 a | 8.5 ± 0.3 b | 8.8 ± 0.1 a | 6.6 ± 0.9 b | 6.6 ± 0.9 b | 7.0 ± 0.4 b | 8.1 ± 0.3 b | 8.1 ± 0.3 b | 8.7 ± 0.1 b |
| | SRF | 11.0 ± 0.6 a | 11.2 ± 0.6 a | 11.7 ± 0.4 a | 9.5 ± 0.5 a | 9.4 ± 0.7 a | 9.6 ± 0.5 a | 7.4 ± 0.4 a | 7.5 ± 0.3 a | 7.7 ± 1.3 a | 9.0 ± 0.5 a | 9.2 ± 0.2 a | 9.7 ± 0.5 a |
| | ARF | 11. 3 ± 0.5 a | 11.5 ± 0.6 a | 12.1 ± 0.3 a | 9.6 ± 0.7 a | 9.7 ± 0.5 a | 9.7 ± 0.3 a | 7.7 ± 0.3 a | 7.7 ± 0.7 a | 7.8 ± 1.2 a | 9.4 ± 0.4 a | 9.4 ± 0.3 a | 9.9 ± 0.4 a |
| TN | CK | 0.76 ± 0.0 c | 0.76 ± 0.0 c | 0.77 ± 0.0 c | 0.47 ± 0.0 b | 0.48 ± 0.0 b | 0.49 ± 0.1 b | 0.40 ± 0.0 b | 0.41 ± 0.0 b | 0.41 ± 0.0 b | 0.53 ± 0.0 b | 0.54 ± 0.0 b | 0.55 ± 0.0 b |
| | FRF | 0.79 ± 0.0 b | 0.80 ± 0.0 b | 0.81 ± 0.0 b | 0.51 ± 0.0 a | 0.52 ± 0.1 b | 0.52 ± 0.1 b | 0.42 ± 0.0 b | 0.43 ± 0.0 b | 0.44 ± 0.0 b | 0.54 ± 0.0 b | 0.56 ± 0.0 b | 0.57 ± 0.02 |
| | SRF | 0.82 ± 0.0 a | 0.84 ± 0.0 a | 0.84 ± 0.0 a | 0.61 ± 0.03 | 0.63 ± 0.0 a | 0.65 ± 0.0 a | 0.50 ± 0.0 a | 0.51 ± 0.0 a | 0.52 ± 0.0 a | 0.64 ± 0.0 a | 0.66 ± 0.0 a | 0.67 ± 0.0 a |
| | ARF | 0.84 ± 0.0 a | 0.86 ± 0.0 a | 0.89 ± 0.0 a | 0.62 ± 0.0 a | 0.64 ± 0.0 a | 0.66 ± 0.0 a | 0.53 ± 0.0 a | 0.54 ± 0.0 a | 0.54 ± 0.0 a | 0.64 ± 0.0 a | 0.66 ± 0.0 a | 0.70 ± 0.0 a |
| TP | CK | 0.68 ± 0.0 b | 0.69 ± 0.0 b | 0.70 ± 0.0 b | 0.65 ± 0.0 b | 0.66 ± 0.0 b | 0.67 ± 0.0 b | 0.63 ± 0.0 b | 0.64 ± 0.0 b | 0.65 ± 0.0 b | 0.65 ± 0.0 b | 0.66 ± 0.0 b | 0.65 ± 0.0 b |
| | FRF | 0.69 ± 0.0 b | 0.71 ± 0.0 b | 0.74 ± 0.0 a | 0.66 ± 0.0 a | 0.67 ± 0.0 b | 0.70 ± 0.0 a | 0.65 ± 0.0 a | 0.67 ± 0.0 b | 0.70 ± 0.0 a | 0.67 ± 0.0 a | 0.68 ± 0.06 | 0.72 ± 0.0 b |
| | SRF | 0.78 ± 0.0 a | 0.79 ± 0.0 a | 0.82 ± 0.0 a | 0.70 ± 0.0 a | 0.72 ± 0.0 a | 0.75 ± 0.0 a | 0.67 ± 0.0 a | 0.69 ± 0.0 a | 0.72 ± 0.0 a | 0.71 ± 0.0 a | 0.73 ± 0.0 a | 0.76 ± 0.0 a |
| | ARF | 0.78 ± 0.0 a | 0.80 ± 0.0 a | 0.84 ± 0.0 a | 0.71 ± 0.0 a | 0.73 ± 0.0 a | 0.76 ± 0.0 a | 0.68 ± 0.0 a | 0.70 ± 0.0 a | 0.73 ± 0.0 a | 0.72 ± 0.0 a | 0.74 ± 0.0 a | 0.77 ± 0.02 |
| TK | CK | 16.5 ± 0.1 b | 16.5 ± 0.1 b | 16.5 ± 0.1 b | 13.7 ± 0.9 b | 13.7 ± 0.2 b | 13.8 ± 0.1 b | 12.5 ± 0.8 b | 12.4 ± 0.1 b | 12.6 ± 0.1 b | 14.5 ± 0.9 b | 14.5 ± 0.2 b | 14.6 ± 0.1 b |
| | FRF | 16.6 ± 0.2 b | 17.2 ± 0.2 b | 17.6 ± 0.2 b | 14.6 ± 0.6 b | 14.8 ± 0.2 b | 15.3 ± 0.2 a | 13.8 ± 0.4 b | 13.6 ± 0.1 b | 14.0 ± 0.2 b | 15.7 ± 0.4 b | 15.5 ± 0.2 b | 16.4 ± 0.2 b |
| | SRF | 18.7 ± 0.5 a | 21.2 ± 1.1 a | 21.7 ± 1.1 a | 16.7 ± 0.9 a | 17.4 ± 0.9 a | 17.7 ± 0.9 a | 15.2 ± 0.8 a | 15.7 ± 0.8 a | 16.1 ± 0.8 a | 17.4 ± 0.9 a | 18.1 ± 0.9 a | 18.5 ± 1.0 a |
| | ARF | 18.9 ± 0.5 a | 21.3 ± 1.1 a | 21.8 ± 1.1 a | 16.8 ± 0.9 a | 17.5 ± 0.9 a | 17.9 ± 0.9 a | 15.3 ± 0.8 a | 15.9 ± 0.8 a | 16.3 ± 0.8 a | 17.6 ± 0.9 a | 18.2 ± 0.9 a | 18.6 ± 1.0 a |
| LOC | CK | 0.74 ± 0.0 c | 0.75 ± 0.0 c | 0.75 ± 0.0 c | 0.58 ± 0.0 d | 0.58 ± 0.0 b | 0.59 ± 0.1 c | 0.46 ± 0.0 b | 0.47 ± 0.0 c | 0.47 ± 0.1 c | 0.56 ± 0.0 b | 0.57 ± 0.0 b | 0.57 ± 0.0 c |
| | FRF | 0.78 ± 0.0 c | 0.79 ± 0.0 c | 0.83 ± 0.1 b | 0.61 ± 0.0 c | 0.64 ± 0.0 c | 0.65 ± 0.0 b | 0.48 ± 0.0 b | 0.47 ± 0.0 b | 0.52 ± 0.0 b | 0.60 ± 0.1 a | 0.62 ± 0.0 a | 0.64 ± 0.1 b |
| | SRF | 0.85 ± 0.1 b | 0.87 ± 0.16 | 0.91 ± 0.1 a | 0.63 ± 0.0 a | 0.68 ± 0.1 a | 0.70 ± 0.0 a | 0.53 ± 0.1 a | 0.52 ± 0.1 a | 0.57 ± 0.1 a | 0.68 ± 0.0 a | 0.69 ± 0.0 a | 0.72 ± 0.0 a |
| | ARF | 0.95 ± 0.1 a | 0.96 ± 0.0 a | 1.01 ± 0.7 a | 0.70 ± 0.0 a | 0.70 ± 0.0 a | 0.74 ± 0.0 a | 0.58 ± 0.0 a | 0.58 ± 0.0 a | 0.62 ± 0.0 a | 0.69 ± 0.0 a | 0.72 ± 0.0 a | 0.74 ± 0.0 a |
| AN | CK | 67.4 ± 0.7 c | 66.2 ± 0.9 d | 67.6 ± 0.9 d | 55.5 ± 1.1 b | 55.0 ± 1.6 c | 56.2 ± 1.6 b | 47.1 ± 0.8 b | 47.4 ± 0.7 b | 48.3 ± 0.8 b | 58.7 ± 0.6 b | 61.3 ± 0.6 b | 62.5 ± 0.7 b |
| | FRF | 71.4 ± 1.2 b | 74.9 ± 1.3 c | 76.3 ± 1.3 c | 55.9 ± 0.3 b | 56.2 ± 1.7 c | 57.4 ± 1.8 b | 46.5 ± 4.6 b | 48.5 ± 4.8 b | 49.4 ± 4.9 b | 60.8 ± 1.0 b | 63.4 ± 1.0 a | 64.6 ± 1.1 a |
| | SRF | 111.6 ± 2.7 a | 116.7 ± 2.8 b | 118.7 ± 2.9 b | 75.6 ± 1.1 a | 77.9 ± 1.1 b | 79.2 ± 1.2 a | 62.6 ± 4.5 a | 65.5 ± 4.7 a | 66.6 ± 4.8 a | 71.0 ± 2.2 a | 74.3 ± 2.3 a | 75.6 ± 2.4 a |
| | ARF | 112.8 ± 2.8 a | 122.4 ± 3.1 a | 124.6 ± 3.2 a | 77.4 ± 2.3 a | 83.0 ± 4.4 a | 84.1 ± 4.6 a | 65.6 ± 0.6 a | 69.7 ± 2.3 a | 70.9 ± 2.4 a | 73.3 ± 1.7 a | 76.7 ± 1.7 a | 78.0 ± 1.8 a |
| AP | CK | 17.9 ± 0.4 c | 18.1 ± 0.4 c | 18.3 ± 0.4 b | 17.2 ± 0.7 c | 17.4 ± 0.7 c | 17.5 ± 0.7 b | 15.3 ± 0.4 c | 15.5 ± 0.5 c | 15.6 ± 0.5 b | 16.6 ± 0.7 c | 16.8 ± 0.8 d | 16.9 ± 0.8 c |
| | FRF | 18.8 ± 0.1 c | 19.0 ± 0.1 c | 19.2 ± 0.1 b | 17.9 ± 0.7 c | 18.0 ± 0.7 c | 18.2 ± 0.7 b | 16.3 ± 0.7 b | 16.4 ± 0.7 b | 16.6 ± 0.7 b | 17.9 ± 0.3 b | 18.1 ± 0.3 c | 18.2 ± 0.3 b |
| | SRF | 27.3 ± 1.3 b | 28.5 ± 1.4 b | 32.3 ± 0.5 a | 26.3 ± 0.9 b | 27.4 ± 1.0 b | 31.1 ± 0.6 a | 23.6 ± 0.3 a | 24.6 ± 0.3 a | 27.9 ± 0.8 a | 26.0 ± 0.6 a | 27.1 ± 0.6 b | 30.8 ± 0.5 a |
| | ARF | 29.1 ± 0.3 a | 30.4 ± 0.3 a | 34.5 ± 0.9 a | 28.2 ± 1.2 a | 29.4 ± 1.3 a | 33.3 ± 0.5 a | 24.1 ± 0.3 a | 25.2 ± 0.3 a | 28.5 ± 0.9 a | 27.0 ± 0.2 a | 28.1 ± 0.2 a | 31.9 ± 0.8 a |
| AK | CK | 172.9 ± 1.1 d | 173.4 ± 1.1 d | 174.0 ± 1.0 d | 123.5 ± 1.5 d | 123.9 ± 1.5 d | 124.3 ± 1.5 d | 115.3 ± 1.0 d | 115.6 ± 1.0 d | 116.0 ± 1.0 d | 137.1 ± 1.8 c | 137.5 ± 1.8 c | 138.0 ± 1.8 c |
| | FRF | 176.7 ± 1.1 c | 177.2 ± 1.1 c | 177.8 ± 0.9 c | 128.1 ± 1.8 c | 128.5 ± 1.8 c | 128.9 ± 1.8 c | 120.4 ± 1.4 c | 120.7 ± 1.4 c | 121.2 ± 1.3 c | 140.0 ± 2.5 c | 140.5 ± 2.5 c | 141.0 ± 2.6 c |
| | SRF | 236.1 ± 1.2 b | 251.0 ± 1.2 b | 262.3 ± 1.3 b | 175.1 ± 1.1 b | 186.1 ± 1.1 b | 194.5 ± 1.2 b | 155.9 ± 1.5 b | 165.8 ± 1.6 b | 173.2 ± 1.7 b | 190.1 ± 0.9 b | 202.1 ± 0.9 b | 211.3 ± 0.9 b |
| | ARF | 246.8 ± 1.6 a | 262.4 ± 1.7 a | 274.3 ± 1.8 a | 180.5 ± 2.7 a | 191.9 ± 2.9 a | 200.6 ± 3.0 a | 168.6 ± 1.5 a | 179.2 ± 1.6 a | 188.3 ± 1.6 a | 200.2 ± 0.9 a | 212.9 ± 0.9 a | 222.5 ± 1.0 a |

Note: Pa., parameter; Tr., treatments; CK (empty control), moldboard plow planting without fertilization; FRF, standard film mulched ridge-furrow planting; ARF, autumn film mulched ridge microfurrow rainwater harvesting planting; SRF, spring film mulched ridge microfurrow rainwater harvesting planting; SOC, soil organic carbon; TN, total nitrogen; TP, total phosphorus; TK, total potassium; LOC, labile organic carbon; AN, alkali-hydrolyzable nitrogen; AP, available phosphorus; AK, available potassium. Different letters in the same column indicate significant differences at the 0.05 level of probability.

Table 3. Yield and yield components of potato under different soil management practices.

| Year | Tr. | Tuber Yield (kg·ha⁻¹) | Total Biomass (kg·ha⁻¹) | Fresh Root (kg·ha⁻¹) | WUE (kg mm⁻¹) | Tuberization Characteristics of Potato | | | | | |
| | | | | | | LPR (%) | MPR (%) | SPR (%) | LPW (kg plant⁻¹) | MPW (kg plant⁻¹) | SPW (kg plant⁻¹) |
|---|---|---|---|---|---|---|---|---|---|---|---|
| 2018 | ARF | 34,693.6 ± 1696.3 a | 44,740.2 ± 4095.9 a | 766.4 ± 93.9 a | 102.0 ± 5.0 a | 40.6 ± 0.2 a | 21.8 ± 1.3 a | 37.6 ± 1.1 b | 0.6 ± 0.02 a | 0.24 ± 0.01 a | 0.20 ± 0.01 a |
| | SRF | 32,667.8 ± 1450.5 ab | 38,389.0 ± 951.0 ab | 669.5 ± 108.9 ab | 94.5 ± 4.9 a | 39.8 ± 0.8 a | 21.2 ± 1.7 a | 39.1 ± 1. 2 b | 0.6 ± 0.01 a | 0.23 ± 0.02 a | 0.19 ± 0.02 a |
| | FRF | 30,716.7 ± 2150.2 bc | 36,505.9 ± 4086.5 bc | 532.3 ± 27.3 bc | 77.2 ± 0.4 b | 34.4 ± 0.4 b | 17.9 ± 1.7 b | 47.7 ± 2.0 a | 0.5 ± 0.02 b | 0.20 ± 0.01 b | 0.21 ± 0.01 a |
| | CK | 28,430.3 ± 2503.4 c | 29,926.6 ± 368.7 c | 522.3 ± 27.3 c | 60.0 ± 1.5 c | 34.9 ± 1.4 b | 17.8 ± 1.1 b | 47.3 ± 2.4 a | 0.5 ± 0.01 b | 0.20 ± 0.02 b | 0.20 ± 0.02 a |
| 2019 | ARF | 42,894.2 ± 1993.9 a | 55,816.1 ± 4556.1 a | 950.4 ± 116.4 a | 110.7 ± 5.0 a | 40.3 ± 1.6 a | 20.1 ± 0.9 a | 39.6 ± 2.2 b | 0.7 ± 0.02 a | 0.25 ± 0.01 a | 0.19 ± 0.01 a |
| | SRF | 40,836.8 ± 2616.4 a | 48,708.5 ± 3570.6 a | 736.5 ± 119.8 b | 104.1 ± 4.4 a | 39.6 ± 1.0 a | 19.5 ± 0.9 a | 40.9 ± 1.9 b | 0.6 ± 0.02 a | 0.24 ± 0.01 a | 0.19 ± 0.01 a |
| | FRF | 39,165.5 ± 2419.2 ab | 37,759.1 ± 3014.8 b | 585.6 ± 30.0 bc | 72.7 ± 3.2 b | 36.1 ± 0.8 b | 17.1 ± 1.6 b | 46.8 ± 1.1 a | 0.5 ± 0.01 b | 0.22 ± 0.01 b | 0.20 ± 0.01 a |
| | CK | 35,550.0 ± 1490.8 b | 31,775.8 ± 2691.4 b | 532.7 ± 27.7 c | 56.1 ± 3.9 c | 35.8 ± 0.8 b | 15.9 ± 2.0 b | 48.3 ± 2.3 a | 0.5 ± 0.02 b | 021 ± 0.10 b | 0.20 ± 0.00 a |
| 2020 | ARF | 63,378.1 ± 323.3 a | 87,187.2 ± 6710.7 a | 1149.6 ± 140.8 a | 150.0 ± 6.5 a | 41.0 ± 0.2 a | 21.7 ± 1.5 a | 37.3 ± 0.15 a | 0.7 ± 0.03 a | 0.25 ± 0.01 a | 0.20 ± 0.01 a |
| | SRF | 62,587.0 ± 87.4 a | 77,791.3 ± 5221.6 b | 970.8 ± 157.9 b | 143.1 ± 6.6 a | 40.1 ± 1.1 a | 21.0 ± 1.6 a | 38.9 ± 0.12 b | 0.7 ± 0.01 a | 0.24 ± 0.01 a | 0.19 ± 0.01 a |
| | FRF | 59,288.1 ± 1357.9 b | 66,600.7 ± 1617.0 c | 745.3 ± 38.2 bc | 101.0 ± 4.1 b | 34.4 ± 0.4 b | 17.9 ± 1.7 b | 47.7 ± 0.15 a | 0.5 ± 0.01 b | 0.21 ± 0.01 b | 0.21 ± 0.01 a |
| | CK | 54,316.7 ± 1227.1 c | 44,792.9 ± 1333.3 d | 668.5 ± 33.6 c | 63.6 ± 2.7 c | 34.9 ± 1.4 b | 17.8 ± 1.1 b | 47.3 ± 0.15 a | 0.5 ± 0.10 b | 0.21 ± 0.01 b | 0.20 ± 0.00 a |
| Ave. | ARF | 46,988.6 ± 347.9 a | 62,427.4 ± 4429.4 a | 700.0 ± 85.8 a | 123.1 ± 2.6 a | 39.9 ± 0.7 a | 20.8 ± 0.6 a | 39.3 ± 1.1 b | 0.7 ± 0.2 a | 0.25 ± 0.1 a | 0.20 ± 0.00 b |
| | SRF | 45,363.9 ± 395.3 b | 55,389.8 ± 4004.1 b | 569.1 ± 92.5 b | 116.3 ± 2.8 a | 39.4 ± 0.4 a | 20.3 ± 0.8 a | 40.4 ± 1.1 b | 0.6 ± 0.0 a | 0.24 ± 0.1 a | 0.19 ± 0.00 b |
| | FRF | 43,056.7 ± 182.2 c | 46,074.4 ± 2309.6 c | 443.6 ± 22.7 c | 85.3 ± 1.0 b | 35.0 ± 0.4 b | 17.7 ± 1.3 b | 47.5 ± 0.8 a | 0.5 ± 0.1 b | 0.21 ± 0.1 b | 0.21 ± 0.01 a |
| | CK | 39,432.3 ± 974.2 d | 35,471.2 ± 1197.3 d | 400.4 ± 216 c | 59.9 ± 1.4 c | 34.9 ± 0.7 b | 16.9 ± 1.2 b | 48.1 ± 1.7 a | 0.5 ± 0.1 b | 0.21 ± 0.1 b | 0.20 ± 0.01 ab |

Note: Tr., treatments; Ave., average; CK (empty control), moldboard plow planting without fertilization; FRF, standard film mulched ridge-furrow planting; ARF, autumn film mulched ridge microfurrow rainwater harvesting planting; SRF, spring film mulched ridge microfurrow rainwater harvesting planting. Total biomass including tuber, haulm, and root. WUE, water-use efficiency. LPR, large potato rate; MPR, medium potato rate; SPR, small potato rate; LPW, large potato weight; MPW, medium potato rate; SPW, small potato rate. Different letters in the same column indicate significant differences at the 0.05 level of probability.

**Table 4.** Potato nutrient-use efficiency (NUE$_F$) of nitrogen, phosphorus, and potassium fertilizers under different planting systems.

| Year | Tr. | Fertilizer Partial Factor Productivity, PFP | | | Agronomic Efficiency, AE | | | Recovery Efficiency, RE | | |
|---|---|---|---|---|---|---|---|---|---|---|
| | | PFP$_N$ (kg·kg$^{-1}$) | PFP$_P$ (kg·kg$^{-1}$) | PFP$_K$ (kg·kg$^{-1}$) | AE$_N$ (kg·kg$^{-1}$) | AE$_P$ (kg·kg$^{-1}$) | AE$_K$ (kg·kg$^{-1}$) | RE$_N$ (kg·kg$^{-1}$) | RE$_P$ (kg·kg$^{-1}$) | RE$_K$ (kg·kg$^{-1}$) |
| 2018 | ARF | 177.5 ± 8.0 a | 312.2 ± 15.3 a | 183.0 ± 8.9 a | 37.9 ± 6.3 a | 56.4 ± 9.3 a | 33.0 ± 5.5 a | 0.60 ± 0.0 a | 0.21 ± 0.0 a | 0.65 ± 0.0 a |
| | SRF | 167.2 ± 7.4 ab | 294.0 ± 13.1 ab | 172.3 ± 7.7 ab | 25.6 ± 7.3 ab | 38.1 ± 10.9 b | 22.3 ± 6.4 ab | 0.47 ± 0.0 b | 0.17 ± 0.0 a | 0.48 ± 0.0 b |
| | FRF | 157.2 ± 11.0 bc | 276.4 ± 19.3 bc | 159.3 ± 12.2 bc | 13.8 ± 8.4 c | 20.6 ± 12.5 c | 12.1 ± 7.3 b | 0.45 ± 0.1 b | 0.15 ± 0.0 a | 0.41 ± 0.1 b |
| | CK | 145.5 ± 12.8 c | 255.8 ± 22.5 c | 148.8 ± 14.2 c | | | | | | |
| 2019 | ARF | 219.5 ± 10.2 a | 386.0 ± 17.9 a | 226.2 ± 10.5 a | 44.4 ± 3.5 a | 66.1 ± 5.2 a | 38.7 ± 3.0 a | 0.67 ± 0.0 a | 0.25 ± 0.0 a | 0.73 ± 0.0 a |
| | SRF | 209.0 ± 13.4 a | 367.5 ± 23.5 a | 215.4 ± 13.8 a | 32.0 ± 7.1 b | 47.6 ± 10.6 b | 27.9 ± 6.2 b | 0.61 ± 0.1 a | 0.22 ± 0.0 a | 0.62 ± 0.1 ab |
| | FRF | 200.4 ± 12.4 ab | 352.4 ± 21.8 ab | 206.6 ± 12.8 ab | 21.9 ± 6.4 b | 32.5 ± 9.5 b | 19.1 ± 5.6 b | 0.61 ± 0.0 a | 0.21 ± 0.0 a | 0.56 ± 0.0 b |
| | CK | 181.9 ± 7.6 b | 319.9 ± 13.4 b | 187.2 ± 7.9 b | | | | | | |
| 2020 | ARF | 334.3 ± 1.7 a | 570.3 ± 2.9 a | 334.3 ± 1.7 a | 54.8 ± 6.8 a | 81.5 ± 10.1 a | 47.8 ± 5.9 a | 0.95 ± 0.1 a | 0.35 ± 0.0 a | 0.99 ± 0.1 a |
| | SRF | 330.1 ± 0.5 a | 563.2 ± 0.8 a | 330.1 ± 0.5 a | 50.0 ± 7.7 ab | 74.4 ± 11.4 ab | 43.6 ± 6.7 ab | 0.90 ± 0.1 a | 0.33 ± 0.0 a | 0.93 ± 0.1 ab |
| | FRF | 312.7 ± 7.2 b | 533.5 ± 12.2 b | 312.7 ± 7.2 b | 30.1 ± 15.5 b | 44.7 ± 23.1 b | 26.2 ± 13.5 b | 0.77 ± 0.1 a | 0.27 ± 0.1 a | 0.73 ± 0.2 b |
| | CK | 286.1 ± 6.5 c | 488.8 ± 11.0 c | 286.1 ± 6.5 c | | | | | | |
| Average | ARF | 240.5 ± 1.8 a | 422.8 ± 3.1 a | 247.8 ± 1.8 a | 45.7 ± 4.0 a | 68.0 ± 5.9 a | 39.9 ± 3.5 a | 0.74 ± 0.1 a | 0.27 ± 0.0 a | 0.79 ± 0.0 a |
| | SRF | 232.1 ± 2.0 b | 408.2 ± 3.6 b | 239.2 ± 2.1 b | 35.9 ± 6.8 ab | 53.4 ± 10.1 ab | 31.3 ± 5.9 ab | 0.66 ± 0.0 ab | 0.24 ± 0.0 ab | 0.68 ± 0.1 ab |
| | FRF | 220.3 ± 0.9 c | 387.4 ± 1.6 c | 226.2 ± 1.1 c | 21.9 ± 6.9 b | 32.6 ± 10.3 ab | 19.1 ± 6.0 b | 0.61 ± 0.1 b | 0.21 ± 0.0 b | 0.57 ± 0.1 b |
| | CK | 201.8 ± 5.0 d | 354.8 ± 8.8 d | 207.4 ± 5.6 d | | | | | | |

Note: Tr., treatment; PFP$_N$, PFP$_P$, and PFP$_K$ represent the partial factor productivity of N, P$_2$O$_5$, and K$_2$O applied, respectively; AE$_N$, AE$_P$, and AE$_K$ represent the agronomic efficiency of N, P$_2$O$_5$, and K$_2$O applied, respectively; RE$_N$, RE$_P$, and RE$_K$ represent the recovery efficiency of N, P$_2$O$_5$, and K$_2$O applied, respectively. The applied nutrients included those from farmyard manure. Different letters in the same column indicate significant differences at the 0.05 level of probability.

| Year | Tr. | Internal Use Efficiency, IE | | | Reciprocal Internal Use Efficiency (RIE) | | |
|---|---|---|---|---|---|---|---|
| | | IE$_N$ (kg·kg$^{-1}$) | IE$_P$ (kg·kg$^{-1}$) | IE$_K$ (kg·kg$^{-1}$) | RIE$_N$ (kg·t$^{-1}$) | RIE$_P$ (kg·t$^{-1}$) | RIE$_K$ (kg·t$^{-1}$) |
| 2018 | ARF | 189.0 ± 8.5 a | 1306.0 ± 96.8 a | 212.8 ± 19.9 a | 5.3 ± 0.2 a | 0.8 ± 0.1 a | 4.7 ± 0.4 a |
| | SRF | 229.9 ± 17.9 a | 1632.9 ± 188.3 a | 271.0 ± 39.3 a | 4.4 ± 0.3 a | 0.6 ± 0.1 a | 3.4 ± 0.5 a |
| | FRF | 228.6 ± 51.2 a | 1773.0 ± 440.3 a | 312.9 ± 83.6 a | 4.5 ± 1.0 a | 0.6 ± 0.1 a | 3.4 ± 0.9 a |
| | CK | | | | | | |
| 2019 | ARF | 212.0 ± 22.3 a | 1442.7 ± 110.7 a | 232.7 ± 14.0 a | 4.7 ± 0.5 a | 0.7 ± 0.1 a | 4.3 ± 0.3 a |
| | SRF | 223.5 ± 43.2 a | 1588.2 ± 264.9 a | 263.6 ± 40.7 a | 4.6 ± 1.0 a | 0.6 ± 0.1 a | 3.9 ± 0.7 a |
| | FRF | 212.4 ± 19.5 a | 1596.6 ± 83.1 a | 275.4 ± 7.5 a | 4.7 ± 0.4 a | 0.6 ± 0.0 a | 3.6 ± 0.1 a |
| | CK | | | | | | |
| 2020 | ARF | 221.8 ± 27.6 a | 1547.4 ± 141.9 a | 253.7 ± 18.6 a | 4.6 ± 0.6 a | 0.7 ± 0.1 a | 4.0 ± 0.3 a |
| | SRF | 230.4 ± 21.9 a | 1621.7 ± 115.0 a | 267.3 ± 16.2 a | 4.4 ± 0.4 a | 0.6 ± 0.0 a | 3.7 ± 0.2 a |
| | FRF | 254.6 ± 32.7 a | 1919.8 ± 332.1 a | 332.0 ± 67.9 a | 4.0 ± 0.5 a | 0.5 ± 0.1 a | 3.1 ± 0.7 a |
| | CK | | | | | | |
| Average | ARF | 207.6 ± 12.4 a | 1432.0 ± 58.4 a | 233.0 ± 7.8 b | 4.9 ± 0.3 a | 0.7 ± 0.0 a | 4.3 ± 0.1 a |
| | SRF | 227.9 ± 14.5 a | 1614.3 ± 124.8 a | 267.3 ± 24.7 ab | 4.4 ± 0.3 a | 0.6 ± 0.1 ab | 3.7 ± 0.2 b |
| | FRF | 231.9 ± 28.4 a | 1763.1 ± 255.3 a | 306.8 ± 49.4 a | 4.4 ± 0.5 a | 0.6 ± 0.1 b | 3.4 ± 0.5 b |
| | CK | | | | | | |

Note: Tr., treatment; IE$_N$, IE$_P$, and IE$_K$ represent the internal use efficiency of N, P$_2$O$_5$, and K$_2$O applied, respectively; RIE$_N$, RIE$_P$, and RIE$_K$ represent reciprocal internal use efficiency, which is the nutrient uptake per ton tube. The applied nutrients included those from farmyard manure. Different letters in the same column indicate significant differences at the 0.05 level of probability.

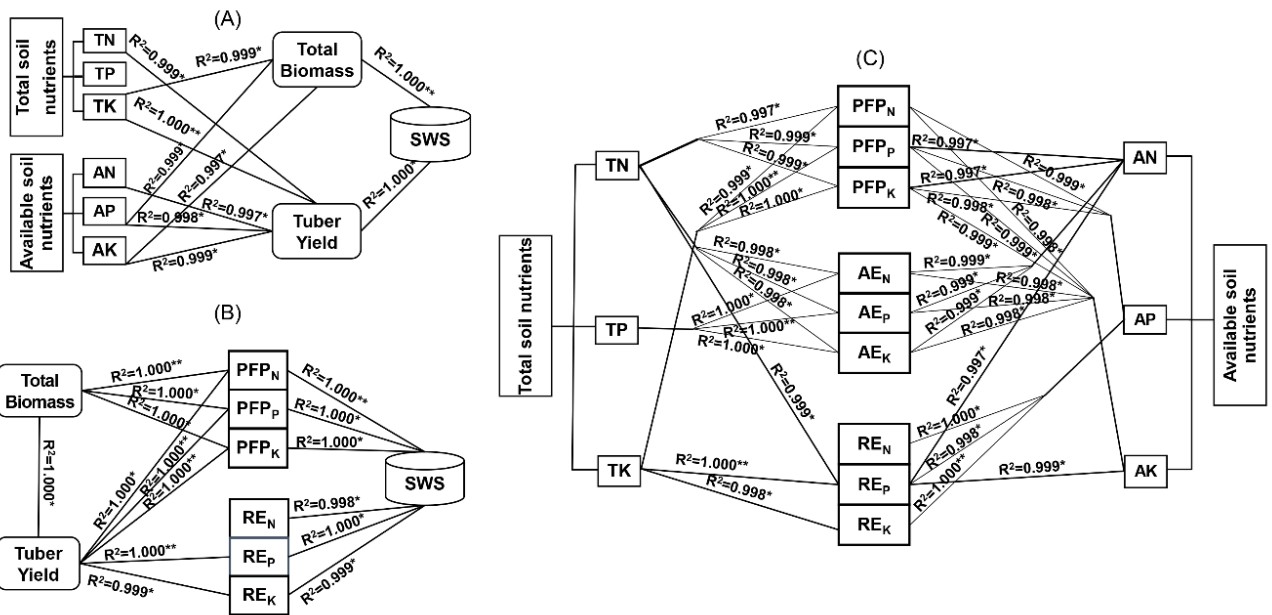

**Figure 3.** Relationship between soil fertility traits, soil water storage, potato tuber and biomass yield, and potato nutrient-use efficiency (NUE$_F$). SOC, TN, TP, and TK and LOC, AN, AP, and AK represent the total and available nutrients, respectively. PFP$_N$, PFP$_P$, and PFP$_K$; AE$_N$, AE$_P$, and AE$_K$; RE$_N$, RE$_P$, and RE$_K$; and IE$_N$, IE$_P$, and IE$_K$ represent partial factor productivity, agronomic efficiency, recovery efficiency, and internal use efficiency of N, P$_2$O$_5$, and K$_2$O applied, respectively. RIE$_N$, RIE$_P$, and RIE$_K$ represent reciprocal internal use efficiency, i.e., the nutrient uptake recovery efficiency in kg per ton of tubers. (**A**) showed the relationship between soil fertility traits, SWS and potato total biomass and tuber yield, (**B**) showed the contribution of potato total biomass and tuber yield, SWS to PFP and RE of the NPK nutrients used, and (**C**) showed the contribution of the total and available soil NPK nutrients to PFP, AE, and RE of the NPK nutrients used. "*" and "**" indicate significant differences at $p < 0.05$ and $p < 0.01$ levels of probability.

## 4. Discussion

### 4.1. Autumn Mulch Ridge Microfurrow Planting Significantly Improves Soil Water and Nutrient Levels and Availability

Potato productivity in China is limited by water scarcity. Film mulched ridge-furrow tillage can effectively modify the plant hydrothermal microenvironment by significantly increasing the moisture and temperature of the topsoil in the early growing season (sowing–emergence), which is ideal for optimizing the soil moisture and soil temperature throughout the growing season and, in turn, improving potato productivity in this semiarid rainfed region [11,40,72,73]. This partly explains why ARF considerably increased the 0–200 cm soil water content in our study. Additionally, long-term mulch immediately after the end of the rainy season combined with rainwater harvesting through microfurrows on the ridge played an important role in increasing and maintaining the soil water under ARF. This was mainly due to ARF increasing the soil water content through its efficient rainwater harvesting effect while protecting the soil water from loss through evaporation through film mulch. ARF increased the soil water status more efficiently, which was significant compared to the results of several similar previous studies that were conducted in a similar region [39,66,72,74].

As an integrated nutrient management practices, film mulched ridge-furrow tillage significantly affects SOC and soil NPK levels [71,75–77]. In our study, we observed that ARF significantly increased the levels of SOC and NPK, which was mainly attributed to the following: First, ARF significantly improved the water and temperature conditions of the topsoil layer [78], which led to the increased vigorous growth of potatoes and retention of increased organic matter (OM), such as stems, leaves, litter, and roots, to the soil. The

OM then rapidly broke down and released more C and NPK nutrients to the soil, which significantly increased the SOC, TN, TP, and TK levels. Our results confirmed the findings of several similar studies in a similar region [12,15,40,59,62], but were more efficiently. However, the modified soil water content and temperature under ARF enhanced the SOC and nitrogen mineralization [79], which contributed to the increased availability of the AN, AP, and AK and confirmed the findings of Qin [40]. Meanwhile, the hydrothermal gradient formed by the high surface temperature and moisture under ARF also promotes the movement of soil C and nutrients into the upper layer of the soil profile [39], which results in the stratification of SOC and NPK [80] and thus further increases the topsoil nutrient content [81–83]. Second, ARF promotes SOC and NPK accumulation in the topsoil layer via ridge tillage and the enhanced positive interaction between N, P, and K nutrients, which increases the plant nutrient uptake and leaves more nutrients in the topsoil [70,76,84]. In addition, the SOC and TP contents were higher in the 0–20 cm soil layer under ARF than those in CK, which may be explained by a faster and greater transformation from fresh OM to SOM than to SOC owing to different soil water moisture and temperature [85]. The AN content was higher in the 20–40 cm soil layer under ARF, possibly owing to the deep plowing (25–30 cm) accumulating more nitrogen in the deep soil.

The film mulching ridges with microfurrow tillage accelerate the surface migration and aggregation of soil nutrients and greatly activate soil nutrients. ARF was more effective than SRF and FRF in increasing the SOC and NPK nutrient availability, which is consistent with the results of several similar studies [86]. This was possibly due to fallow mulching enhancing the soil maturation process and the associated physicochemical process, accelerating the decomposition of native and added OM, converting more OM into SOC, and releasing more available nutrients into the soil [87,88]. Furthermore, the increased availability of soil nutrients can also be attributed to the establishment of potato root systems induced by the increased P and K availability under ARF [51,52,56], which assists plants to take up more NPK nutrients from the deep soil into the topsoil. In addition, the immobile properties of phosphorus can facilitate an increase in the TP and AP in the topsoil [89], and the potassium-like properties of potatoes also contribute to increasing the AK [44,90].

### 4.2. Autumn Film Mulched Ridge Microfurrow Planting Increases Potato Yield and Nutrient-Use Efficiency Associated with Increasing Soil Water and Nutrient Availability

Apart from using good varieties and improving the soil water conditions, a stable and desirable tuber yield depends on good soil fertility [75,76,91]. Full film mulching ridge-furrow planting increases the potato yield by increasing the potato water-use efficiency in semiarid areas, especially around the 400 mm rainfall areas in Northwest China, which is well documented [11]. This can be partly explained by the fact that the potato yield increased under ARF in the present study. In fact, we observed that the 0–200 cm soil water storage (SWS) significantly correlated with the potato tuber and biomass yield and ARF significantly increased the potato yield and WUE at the same time, which suggests that ARF is of great importance for the development of potato production to ensure food security in China. Besides water shortages, low fertility is the major restriction to increasing potato yield in Northwest China [9,12,18]. Thus, the significant increase in the SOC, total NPK, and availability of NPK nutrients can partly explain the significant increase in potato tuber yield under ARF. In a field study in Gansu, Northwest China, Qin [40] found that complete mulching with ridge planting significantly increased the available NPK and OM content in the soil and thus significantly improved the potato yield. Moreover, similar studies conducted in India [92] and sub-Saharan Africa [91] have shown that optimal soil fertility management can significantly increase potato tuber yield. Our study showed that the TK, AP, and AK were positively and significantly correlated with the potato biomass yield, and the AN, AP, and AK were positively and significantly correlated with the tuber yield. This indicates that better soil fertility management under ARF increases nutrient availability in the soil, provides optimal nutrition to the potato crop, increases the overall performance of ARF cropping systems and thus increases the total biomass and tuber production [35,47]. In

addition, increasing the potato yield under ARF considerably contributes to good vegetative growth, which is enhanced by the improved availability and effectiveness of soil nutrients and results in better photosynthesis, increased photosynthetic products, and consequently improved tuber expansion, which provides good tuber sizes and an increased potato yield. This is consistent with the results of Qin [39] but is more significant. Previous studies have reported that increasing the nutrient content and availability contributes to increasing the potato nitrogen-use efficiency [21,40,52,57,63,93]. In our study, we observed that ARF consistently increased the $NUE_F$ of the applied NPK nutrients not only by improving the soil nutrient levels and availability but also by improving the soil water levels. We observed that water storage (SWS) in the 0–200 cm soil profile was strongly correlated with the potato tuber and biomass yield. This suggests that ARF will alleviate the problem of water shortage in the study area and will greatly help increase potato production. The improved soil water and nutrient environment then significantly increased the $NUE_F$. The calculated PFP, AE, and RE of the NPK nutrients were all higher under ARF than under SRF and FRF, whereas the calculated IE and RIE showed no significant differences between them and CK. Furthermore, the total and available soil nutrients positively correlated with the PFP, AE and RE, which suggests that the soil nutrient content and availability improved the $NUE_F$. In addition, the $NUE_F$ components PFP, AE, and RE of the NPK nutrients varied in close association with the yield differential, which is also related to soil nutrient content and availability [19,68]; the changes in the IE and RIE were possibly related to the same cultivar being used and its nutritional requirements and intake characteristics [30,32]. Our results indicate that integrated soil water and nutrient management can enhance the positive interaction between N, P, and K nutrients that facilitate the plant uptake of NPK nutrients in a balanced way, which also contributes to the increase in the $NUE_F$ and $NUE_F$ components [17,71]. In addition, the SWS was significantly correlated with the PFP and RE; the AN was significantly positively correlated with the PFP and AE of the NPK nutrients; the AP was significantly positively correlated with the PFP, RE, and AK, which were significantly positively correlated with the PFP and AE in our study, which further suggests that increasing the soil water and nutrient availability is of considerable importance in increasing the potato $NUE_F$. Notably, the estimated PFP, AE, IE, and RIE values of the NPK fertilizer in our study were slightly or significantly higher than those reported by Liu [12] and Xu [13] in a similar study in the same area and also varied in similar trends. In addition, according to the concept and interpretation of NUE and its components in some theoretical [35,38] and practical studies [13,37], the calculated values of the $NUE_F$ and its components in our study fell within the high range of well-managed systems and significantly exceeded the results of several similar studies [12,13,59,63], which suggests that ARF is the best planting system to increase potato yield and NUE in dryland farming. In addition, the location, soil type, rainfall, crop variety, tillage regime, fertilizer regime, and amount can have a major impact on the $NUE_F$. Therefore, further studies should fully consider the integrated effect of these factors.

## 5. Conclusions

Autumn film mulch ridge microfurrow rainwater harvesting (ARF) significantly increases soil water and nutrient levels and availability and greatly increases potato biomass, tuber yield, and NPK nutrient-use efficiency ($NUE_F$) in semiarid areas in Northwest China. Compared to SRF, FRF, and CK, ARF most significantly and efficiently increased the soil water content of the soil profile from 0–200 cm and increased the content of the total and available NPK nutrients in the soil layer from 0–40 cm, particularly the LOC, AN, AP, and AK. This consequently significantly increased the potato tuber and biomass (especially the root biomass) yield and $NUE_F$. On average over three years, the ARF-induced potato tuber yield, total biomass, fresh root, and WUE increased by 19.1%, 76.0%, 74.8%, and 105%, respectively, compared to CK. Typically, ARF concomitantly greatly increased the PFP, AE, RE, and RIE of the NPK nutrients used, with the $PFP_N$, $PFP_P$, and $PFP_K$ increasing by 19.2%, 19.1%, and 19.5% compared to CK; the $AE_N$, $AE_P$, $AE_K$, $RE_N$, $RE_P$, $RE_K$, $RIE_N$, $RIE_P$,

and $EIR_K$ reached 45.7 kg·kg$^{-1}$, 68.0 kg·kg$^{-1}$, 39.9 kg·kg$^{-1}$, 0.74 kg·kg$^{-1}$, 0.27 kg·kg$^{-1}$, 0.79 kg·kg$^{-1}$, 4.9 kg·t$^{-1}$, 0.7 kg·t$^{-1}$, and 4.3 kg·t$^{-1}$, respectively. The tested parameters all fell within the upper range of the well-managed systems and significantly exceeded the results of several similar studies. The TN, TP, TK, and especially the AN, AP, AK, and SWS, were highly significantly correlated with the $NUE_F$, which suggests that improved soil water and nutrient availability and their positive interaction were significantly correlated with increases in the potato tuber yield and $NUE_F$. ARF enhanced the potato plant vigor growth, which resulted in the good development of the root system and also contributed more to the improvement in the tuber yield and $NUE_F$. The PFP and AE of the applied NPK nutrients strongly correlated with soil fertility management practices, whereas the RE and IE increasingly correlated with potato variety traits. The calculated PFP, AE, RE, IE, and RIE of the NPK nutrients were all higher than those obtained in similar studies in the same regions of Northwest China and fell within the range of well-managed farming systems. ARF, which balances optimal nutrient-use efficiency and crop productivity and achieves synchrony between high yield and $NUE_F$, is the most efficient planting system for improving the $NUE_F$ and optimizing the potato yield in dryland farming.

**Author Contributions:** Conceptualization, F.Y. and B.H.; methodology, F.Y.; software, F.Y. and B.D.; validation, F.Y., B.H. and G.Z.; formal analysis, F.Y., B.D. and G.Z.; investigation, B.H., B.D. and G.Z.; resources, B.H.; data curation, F.Y.; writing—original draft preparation, F.Y. and B.H.; writing—review and editing, F.Y.; visualization, F.Y. and G.Z.; supervision, F.Y.; project administration, F.Y.; funding acquisition, F.Y. All authors have read and agreed to the published version of the manuscript.

**Funding:** This research was funded by the National Natural Science Foundation of the People's Republic of China, grant numbers 31860131 and 31560137, and the Gansu Science and Technology Project, People's Republic of China, grant number 18YF1NA095-2.

**Data Availability Statement:** Data sharing is not applicable to this article.

**Acknowledgments:** We acknowledge the support from the National Natural Science Foundation of China (Grant Nos. 31860131 and 31560137) and Gansu Science and Technology Project (Grant No. 18YF1NA095-2), China.

**Conflicts of Interest:** The authors declare that they have no known competing financial interests or personal relationships that could have influenced the work reported in this study.

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
