# Peer review of "Autumn Film Mulched Ridge Microfurrow Planting Improves Yield and Nutrient-Use Efficiency of Potatoes in Dryland Farming"

_agronomy, doi:10.3390/agronomy13061563_

Round 1

Reviewer 1 Report

I think this is an interesting study with a few small points that need attention.

1.Lines 183-184: Direct baking of soil? Or in what container?

2. Specific data should be added to the conclusion section.

3. Discussion is to be strengthened and it should compare similar studies and how this study is better than the previous ones must be clearly indicated by the authors.

4.  A thorough scientific editing is mandatory for the manuscript:Line 433, the large blank space between "ARF" and "is" should be removed, and in line 524, "Acknowledgements:" all need to be bold.

Additionally, I believe that the language of this manuscript needs further revision and refinement.

Reviewer 2 Report

I have suggested some changes in the attached PDF, which are self-explanatory.

The article is well-planned and well-written.

The English language is acceptable.

Author Response

Please see th attachment
